# Endothelial cells express NKG2D ligands and desensitize antitumor NK responses

Thornton W Thompson, Alexander Byungsuk Kim, P Jonathan Li, Jiaxi Wang, Benjamin T Jackson, Kristen Ting Hui Huang, Lily Zhang, David H Raulet*

Department of Molecular and Cell Biology, Cancer Research Laboratory, University of California, Berkeley, United States

**Abstract** Natural Killer (NK) cells confer protection from tumors and infections by releasing cytotoxic granules and pro-inflammatory cytokines upon recognition of diseased cells. The responsiveness of NK cells to acute stimulation is dynamically tuned by steady-state receptor-ligand interactions of an NK cell with its cellular environment. Here, we demonstrate that in healthy WT mice the NK activating receptor NKG2D is engaged in vivo by one of its ligands, RAE-1ε, which is expressed constitutively by lymph node endothelial cells and highly induced on tumor-associated endothelium. This interaction causes internalization of NKG2D from the NK cell surface and transmits an NK-intrinsic signal that desensitizes NK cell responses globally to acute stimulation, resulting in impaired NK antitumor responses in vivo.

DOI: https://doi.org/10.7554/eLife.30881.001

## Introduction

Natural Killer (NK) cells are key effectors in the immune response to pathogens and tumors (*Vivier et al., 2008*). NK cells respond to infected or transformed cells by releasing cytotoxic granules and anti-tumor cytokines such as interferon-γ (IFNγ) (*Vivier et al., 2008*; *Marcus et al., 2014*). NK cells recognize unhealthy cells using an array of cell surface receptors (*Vivier et al., 2011*; *Marcus et al., 2014*; *Moretta et al., 2014*; *Morvan and Lanier, 2016*). These receptors transmit activating or inhibitory signals upon binding cognate ligands on the target cell, and the net balance of these signals dictates whether the NK cell response is triggered. Tumors are often recognized and killed by NK cells in vitro and in vivo because cancer cells tend to upregulate ligands for activating receptors and downregulate ligands for inhibitory receptors (*Waldhauer and Steinle, 2008*; *Marcus et al., 2014*).

The responsiveness of NK cells to a given stimulus is dynamically tuned by the steady-state receptor-ligand interactions experienced by the NK cells (*Joncker and Raulet, 2008*; *Brodin et al., 2009*; *Joncker et al., 2009*; *Joncker et al., 2010*; *Shifrin et al., 2014*). Increases in steady-state stimulation cause NK cells to compensate by adopting a less responsive state (*Joncker et al., 2010*; *Kadri et al., 2016*) – a process that will be referred to here as 'desensitization' – whereas NK cells receiving lower steady-state levels of stimulation exhibit a state of heightened responsiveness to acute activation. For example, the Ly49 family of inhibitory receptors on NK cells are known to engage host MHC I molecules at steady state, and this interaction is important for regulating NK responsiveness. Mice lacking MHC I molecules or inhibitory Ly49 receptors show dramatically weaker NK responses to a wide variety of acute stimulatory signals in vitro and in vivo (*Liao et al., 1991*; *Fernandez et al., 2005*; *Kim et al., 2005*; *Anfossi et al., 2006*; *Brodin et al., 2009*; *Joncker et al., 2010*).

Desensitization may prevent NK cells from effecting autoreactivity and enable them to adjust to different tissue milieus, and mature NK cells can alter their responsiveness upon encountering a new MHC I environment (*Joncker and Raulet, 2008*; *Elliott et al., 2010*; *Joncker et al., 2010*; *Narni-*

*For correspondence:
raulet@berkeley.edu

**eLife digest** White blood cells called "natural killer cells" are part of the first line of immune defense. Often called NK cells for short, one job of these cells is to help prevent cancer by killing tumor cells. If an NK cell spots a tumor cell, it must become energized so that it can deliver the killing blow, which comes in the form of a packet of cell-killing "cytotoxic" granules. Yet tumor cells look very similar to healthy cells, and NK cells must be able to tell the difference to be effective.

Molecules on the outer surface of the NK cell control how the cell recognizes tumors, and deliver the signals the cell needs to become energized. One of these surface molecules is called NKG2D. It interacts with "partner" molecules found on the surface of cancer cells and tells the NK cell to attack. These partner molecules are not usually found on healthy cells, helping the immune system to tell the difference.

After NKG2D interacts with its partner molecules, it moves inside the NK cell. This makes the cell less able to become energized. If the NK cells do not encounter any partner molecules in healthy mice, blocking the interactions should have no effect on NKG2D levels. But now, Thompson et al. find that blocking one of these interactions increased the levels of NKG2D on the surface of NK cells in healthy mice. Further experiments revealed that NK cells in mice constantly encounter an NKG2D partner molecule called RAE-1ε.

A search for the source of RAE-1ε in healthy mice pointed to blood vessels inside the lymph nodes. NK cells pass through theses organs as part of their normal path around the body. Thompson et al. also saw that NK cells from healthy mice were less responsive than NK cells from mutant mice that lacked RAE-1ε. As a result of their encounters with RAE-1ε in healthy mice, the NK cells were less able to kill tumor cells. Blocking the interaction between NKG2D and RAE-1ε in mice re-energized their NK cells. More cells were able to enter tumors in these mice and the cells became better at killing tumors.

Together these findings increase the current understanding of the biological processes that control NK cells. Further research may lead to new treatments for diseases like cancer. But first, scientists need to find out whether NK cells behave in the same way in humans as they do in mice. If so, developing ways to block the interaction could re-energize human NK cells to better kill cancer cells.

DOI: https://doi.org/10.7554/eLife.30881.002

Mancinelli et al., 2013). These dynamics are relevant for antitumor responses, as NK cells in WT mice become desensitized when they infiltrate MHC I-deficient tumors but not when they infiltrate matched MHC-I-positive tumors (Ardolino et al., 2014). Similarly, humans receiving HLA-mismatched bone marrow show altered NK responses that match the trends described in mice (Boudreau et al., 2016).

It is presumed that steady-state interactions between MHC I and Ly49 receptors prevent NK desensitization by inhibiting steady-state signals from activating receptors. Indeed, transgenic over-expression of NK activating ligands causes NK desensitization (Oppenheim et al., 2005; Wiemann et al., 2005; Sun and Lanier, 2008; Tripathy et al., 2008), but the endogenous receptor-ligand systems that transmit these activating signals to NK cells in healthy WT animals remain incompletely defined. In humans, activating KIR appear to be one such endogenous signal involved in steady-state NK cell tuning (Fauriat et al., 2010). In mice, the activating receptor NKp46 may contribute to NK desensitization because NKp46-KO animals showed heightened NK responses to stimulation in one report (Narni-Mancinelli et al., 2012), although not in another (Sheppard et al., 2013). SLAM receptors are also reported to regulate NK responsiveness in some contexts (Chen et al., 2016) (Veillette, 2010).

Very little is understood about which host cell types are responsible for engaging NK cells to regulate responsiveness. A recent study using β2M-KO bone marrow chimeras suggested that MHC-I-deficient nonhematopoietic cells may play a larger role than MHC-I-deficient hematopoietic cells in desensitizing NK cells, although both may participate (Shifrin et al., 2016). In humans, different studies have implicated HLA molecules on hematopoietic cells (Haas et al., 2011) and nonhematopoietic cells (Cooley et al., 2011) as being critical for tuning. Clearly, much remains to be learned

about these processes. Elucidating the receptor-ligand and cellular systems that regulate NK responses in homeostasis and cancer may suggest novel therapeutic strategies.

NKG2D is a C-type lectin-like activating receptor expressed by all NK cells and subsets of T cells (*Raulet, 2003*). NKG2D binds a diverse array of MHC-like proteins. In mice, these include the RAE-1 family (with α, β, γ, δ, and ε isoforms), the H60 family (a, b, c), and MULT1. Human NKG2D ligands include the ULBP family (with isoforms 1–6) and the MICA and MICB proteins (*Raulet et al., 2013*). Acute NKG2D engagement transmits powerful activating signals through the adaptor molecules DAP10 and DAP12 to drive cytotoxicity and cytokine production (*Raulet, 2003*). NKG2D ligands are thought to be absent from most healthy cells but can be induced consequent to DNA damage, oncogene signaling, and other stresses associated with cancer and infection (*Raulet et al., 2013*). Many tumor cells express NKG2D ligands. In tumor transplant and spontaneous cancer models, expression of NKG2D ligand(s) on tumor cells triggers NK activation and protects the host from cancer (*Diefenbach et al., 2001*; *Guerra et al., 2008*).

Interestingly, several recent studies have shown that NK cells in NKG2D-KO mice are hyper-responsive to stimulation when triggered through other activating receptors (*Zafirova et al., 2009*; *Sheppard et al., 2013*). Furthermore, tumor cells engineered to secrete soluble monomeric NKG2D ligands – which block but do not activate NKG2D – increase the responsiveness of tumor-infiltrating NK cells and enhance tumor rejection (*Deng et al., 2015*). These data suggest that NKG2D may contribute to NK desensitization at steady state or in tumors.

In this report, we provide important new findings concerning the cells and molecules that engage NK cells and regulate NK responsiveness, and we clarify the pleiotropic effect of NKG2D on NK activity. Unexpectedly, we show a steady-state interaction between NKG2D and one of its ligands, RAE-1ε, in healthy WT mice. Using bone marrow chimera experiments, we show that non-hematopoietic cells are the primary source of endogenous RAE-1ε. Endothelial cells in lymph nodes were found to be constitutively express RAE-1ε, and RAE-1ε was found to be super-induced on tumor-associated vasculature in transplant and autochthonous cancer models. Importantly, we demonstrate that this interaction between NKG2D and endogenous RAE-1ε desensitizes NK cells and impairs antitumor NK responses and tumor rejection.

## Results

### NKG2D is constitutively engaged by endogenous RAE-1ε

Cell surface NKG2D ligand expression is usually considered a hallmark of unhealthy cells, but expression on the surface of normal cells in healthy animals has not been exhaustively surveyed in vivo. NKG2D is known to be internalized upon ligand engagement (*Lanier, 2015*), so we reasoned that if NKG2D ligands are expressed and interact with NKG2D in healthy WT mice, antibody blockade of the relevant ligand(s) should result in increased levels of NKG2D on the surface of NK cells. Adult C57BL/6 (B6) mice were injected with confirmed blocking antibodies (*Figure 1—figure supplement 1A*) specific for NKG2D ligands RAE-1δ, RAE-1ε, or MULT1. NKG2D levels on NK cells were analyzed by flow cytometry 48 hr post-injection. In vivo blockade of RAE-1ε, but not RAE-1δ or MULT1, substantially increased NKG2D surface levels on NK cells in blood (*Figure 1A*), lymph nodes, and spleen (*Figure 1—figure supplement 1B*). NKG2D elevation after RAE-1ε blockade occurred as early as 12 hr after antibody injection (*Figure 1B*). We subsequently analyzed NKG2D surface levels in RAE-1-KO mice, which contain frameshift mutations (induced by CRISPR/Cas9) in the genes for both RAE-1ε and RAE-1δ (*Deng et al., 2015*). In healthy, unmanipulated animals, NK cells in RAE-1-KO mice showed substantially higher cell surface NKG2D levels than WT controls in all compartments tested, including blood, spleen, lymph nodes, and peritoneal wash (*Figure 1C*). NK cells in bone marrow and liver also showed elevated NKG2D levels in RAE-1-KO mice (*Figure 1—figure supplement 2A*). mRNA levels for *Klrk1* (the gene for NKG2D) were identical in NK cells from WT and RAE-1-KO mice (*Figure 1D*), consistent with the conclusion that host RAE-1ε causes internalization of NKG2D from the NK cell surface. Blocking RAE-1ε in WT mice increased NKG2D to levels comparable to RAE-1-KO mice at steady state, whereas anti-RAE-1ε had no effect on NKG2D levels in RAE-1-KO mice (*Figure 1—figure supplement 1C*). Furthermore, blockade of RAE-1ε in combination with RAE-1δ in WT mice showed no additional effect on NKG2D levels compared with blocking RAE-1ε alone (*Figure 1—figure supplement 1D*).

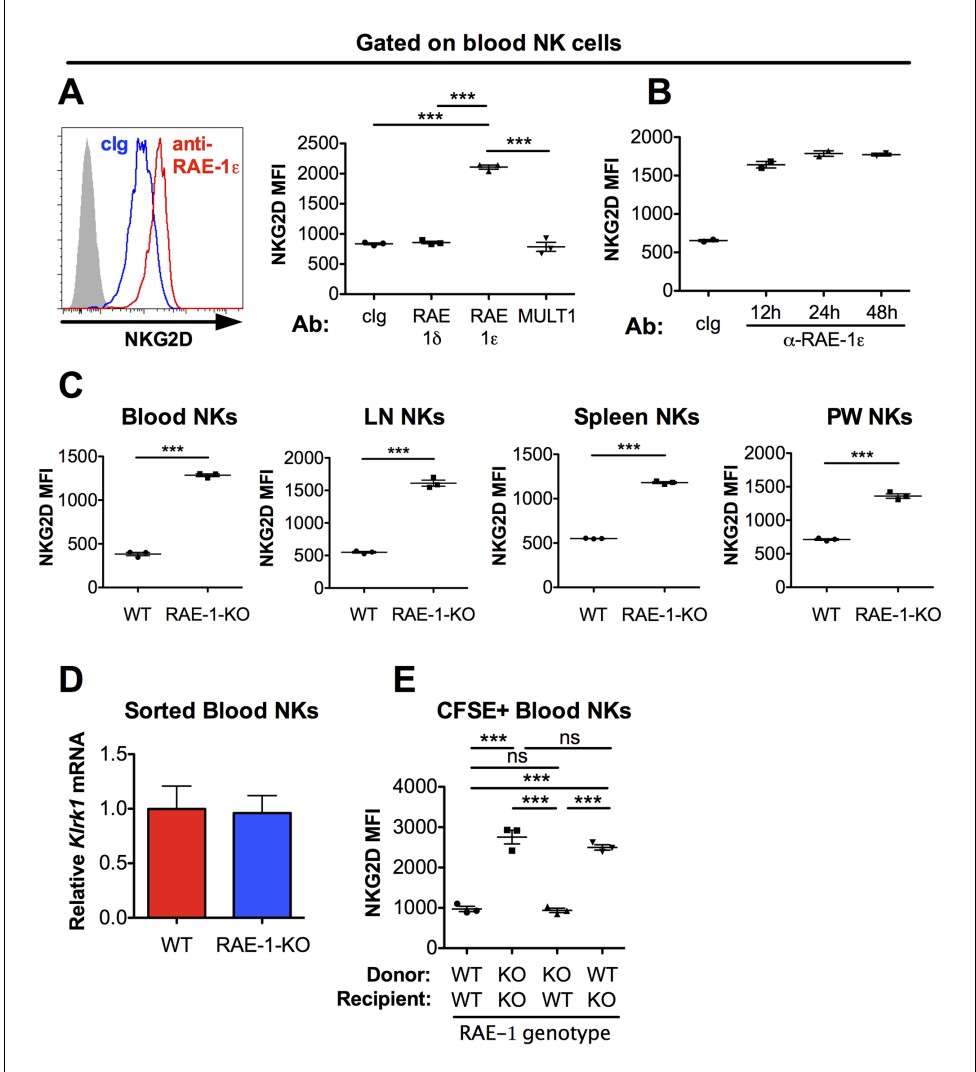

**Figure 1.** NKG2D is engaged and internalized by constitutive interactions with endogenous RAE-1ε in vivo. (A) NKG2D surface levels measured by flow cytometry of blood NK cells 48 hr after injection of blocking antibody specific for the indicated NKG2D ligand. Data are representative of >4 independent experiments. (B) NKG2D surface levels on blood NK cells analyzed at the indicated time point after injection of anti-RAE-1ε. Data are representative of two independent experiments. (C) NKG2D surface levels on blood, lymph node, spleen, and peritoneal wash NK cells in RAE-1-KO mice or WT controls at steady state. Data are representative of >4 independent experiments. (D) Relative *Klrk1* mRNA levels in blood NK cells sorted from WT or RAE-1-KO mice (n = 3) as measured by qRT-PCR. Data are representative of two independent experiments. (E) NKG2D surface levels on CFSE-labeled blood NK cells 48 hr after splenocyte transfer between WT and RAE-1-KO mice. Data are representative of two independent experiments. Statistical significance was determined using one-way ANOVA with Bonferroni post-tests (A, E) or a two-tailed unpaired Student's t tests (C). Data represent means ± SEM.

DOI: https://doi.org/10.7554/eLife.30881.003

The following figure supplements are available for figure 1:

**Figure supplement 1.** Blockade of RAE-1ε results in NKG2D upregulation.
DOI: https://doi.org/10.7554/eLife.30881.004

**Figure supplement 2.** RAE-1-deficiency results in NKG2D upregulation in NK cells in bone marrow and liver.
DOI: https://doi.org/10.7554/eLife.30881.005

To assess whether these phenotypes were intrinsic to NK cells, we transferred CFSE-labeled splenocytes from WT into RAE-1-KO mice and vice versa. When splenocytes were transferred from WT to RAE-1-KO mice, NKG2D levels on the transferred NK cells increased to match the RAE-1-KO mice (*Figure 1E*). Reciprocally, NKG2D surface levels were reduced on NK cells transferred from RAE-1-KO into WT mice. Cumulatively, these data demonstrated that in healthy WT mice a subset of

cells express RAE-1ε, which engages and downregulates NKG2D at steady state from the surface of NK cells.

## Endogenous RAE-1ε diminishes NK responsiveness

We next sought to understand the effect of host RAE-1ε on the function of NK cells. Splenic NK cell numbers and expression of CD11b and CD27 – cell surface markers associated with NK maturation (*Hayakawa and Smyth, 2006*) – were similar in WT and RAE-1-KO mice (*Figure 2—figure supplement 1A*). Release of cytotoxic granules and IFNγ are important NK cell functions (*Vivier et al., 2008*), so we analyzed these responses in WT and RAE-1-KO NK cells after acute ex vivo activation through a variety of receptors. We used a standard 5 hr responsiveness assay in which cells were stimulated by plate-bound antibodies that crosslink activating NK receptors, followed by flow cytometry for degranulation (marked by CD107a cell surface presentation) and intracellular IFNγ (*Joncker et al., 2009*, *2010*). As is typical with this assay, stimulation through the activating receptor NKp46 triggered robust NK cell degranulation and IFNγ production from WT splenic NK cells, and a significantly greater percentage of NK cells from RAE-1-KO mice responded to stimulation compared with WT NK cells (*Figure 2A and B*). NK cells from RAE-1-KO mice also showed elevated responses when stimulated with platebound antibodies that ligate a distinct activating receptor, NK1.1, or that ligate NKG2D itself (*Figure 2B*). These data indicated that splenic NK cells from RAE-1-KO mice exhibit a hyper-responsive phenotype upon acute stimulation through a variety of activating receptors.

In our experience, NK cells in the peritoneal cavity typically yield relatively low responses to ex vivo stimulation. We tested whether endogenous RAE-1ε regulated the responsiveness of these cells. Interestingly, peritoneal NK cells from RAE-1-KO mice showed markedly greater responses compared with their WT counterparts when stimulated through NKp46, NK1.1 or NKG2D (*Figure 2C*). This especially large increase gave us a greater window to examine the desensitization effect, so we next analyzed peritoneal NK responses after injecting WT mice i.p. with antibodies that block RAE-1ε. Similar to the RAE-1-KO mice, blockade of RAE-1ε caused a substantial increase in NK responses to stimulation through all receptors tested (*Figure 2D*). The increased responses could be seen as early as 48 hr after antibody administration.

To analyze killing of tumor cells, we performed a standard 4 hr $^{51}$Cr in vitro cytotoxicity assay, using YAC-1 cells as targets. Peritoneal wash cells from WT, RAE-1-KO, and NKG2D-KO mice were used as effectors. NKG2D-KO effectors were significantly less efficient at killing YAC-1 cells (*Figure 2—figure supplement 1B*), consistent with published reports showing that NKG2D-mediated recognition is required for efficient YAC-1 killing (*Jamieson et al., 2002*; *Guerra et al., 2008*). In contrast, RAE-1-KO mice showed markedly enhanced NK killing of YAC-1 cells (*Figure 2—figure supplement 1B*). Together, these data suggested that endogenous, steady-state RAE-1ε expression desensitizes NK responses to activation through multiple activating receptors and YAC-1 cells in vitro.

## NKG2D regulates NK responsiveness in a cell-intrinsic manner

RAE-1ε binds NKG2D, so we expected NKG2D-KO NK cells to be hyper-responsive to NKG2D-independent stimuli. Indeed, NKG2D-KO NK cells from spleen and peritoneal wash showed increased responses to stimulation compared with WT controls when stimulated through NKp46 and NK1.1 (*Figure 3A and B*), as has also been previously reported (*Zafirova et al., 2009*; *Sheppard et al., 2013*; *Deng et al., 2015*). We then directly compared the responses of peritoneal NK cells from matched WT, RAE-1-KO, and NKG2D-KO mice. When stimulated with platebound antibody ligating NKG2D, NK cells from RAE-1-KO showed elevated responses, whereas NKG2D-KO NK cells failed to respond, as expected (*Figure 3—figure supplement 1A*). In contrast, stimulation through NKp46 resulted in elevated responses from both the RAE-1-KO and NKG2D-KO cohorts (*Figure 3C*). Interestingly, NKG2D-KO NK cells were consistently even more responsive than the NK cells from RAE-1-KO mice (*Figure 3C*). These data suggested that, in addition to RAE-1ε, other ligands may participate in NKG2D-mediated desensitization, or NKG2D may regulate NK responses partly through a ligand-independent mechanism in addition to the RAE-1ε-dependent mechanism documented herein.

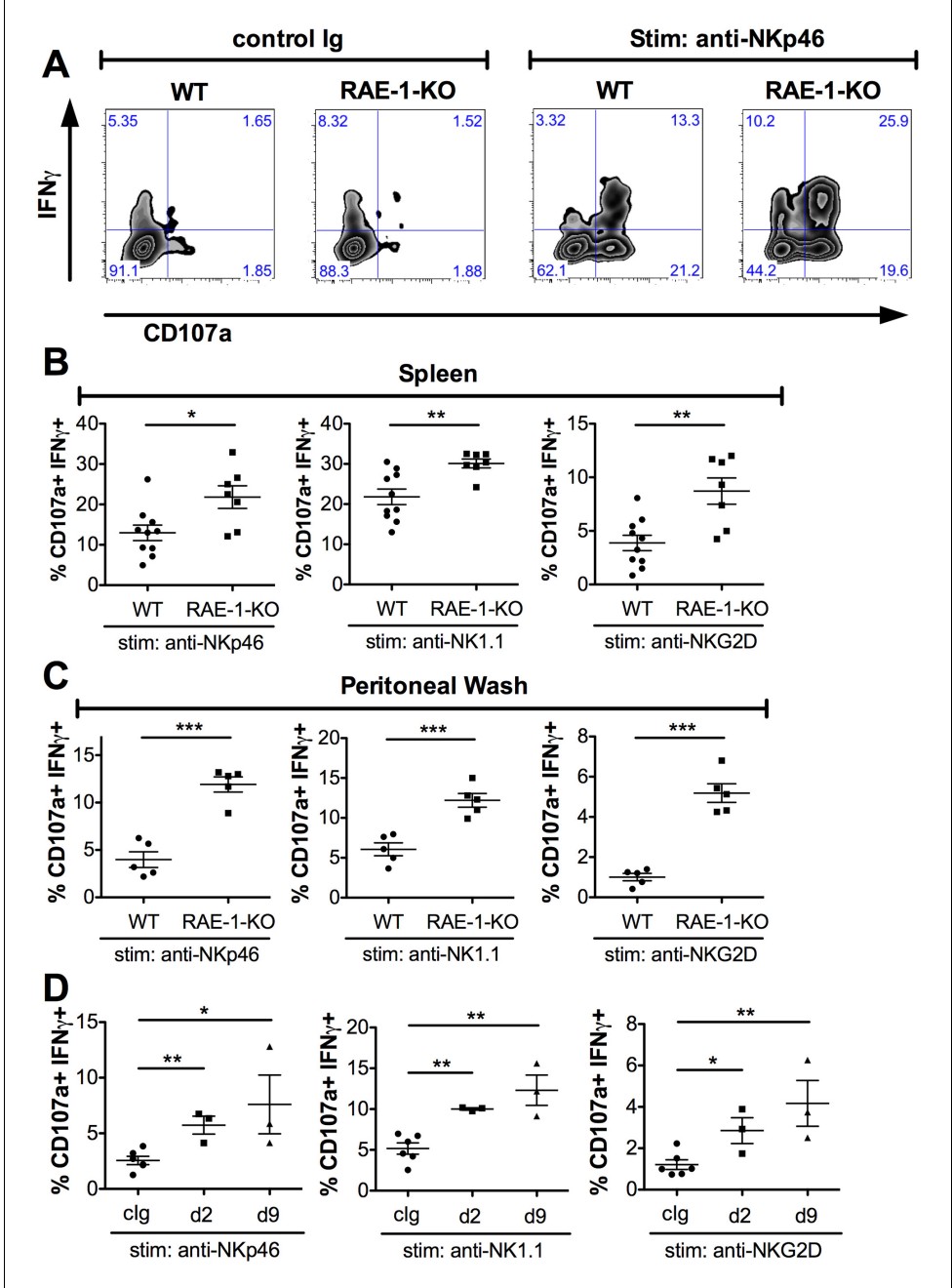

**Figure 2.** Endogenous RAE-1ε negatively regulates NK responsiveness. (**A**) WT or RAE-1-KO splenic NK cell IFNγ production and degranulation (CD107a) after 5 hr ex vivo stimulation with platebound control Ig or anti-NKp46. (**B and C**) Percentage of activated (IFNγ- and CD107a-double-positive) splenic or peritoneal NK cells from WT or RAE-1-KO mice after ex vivo stimulation with the indicated plate-bound antibody. Data are representative of >4 independent experiments. (**D**) Percentage of activated peritoneal NK cells after ex vivo stimulation from mice given control Ig or anti-RAE-1ε for the indicated time. Data are representative of two independent experiments. Statistical significance was determined using two-tailed unpaired Student's t tests. Data represent means ± SEM.

DOI: https://doi.org/10.7554/eLife.30881.006

The following figure supplement is available for figure 2:

**Figure supplement 1.** Normal NK cell cellularity and differentiation but enhanced NK-mediated tumor cell killing in RAE-1-knockout mice.
DOI: https://doi.org/10.7554/eLife.30881.007

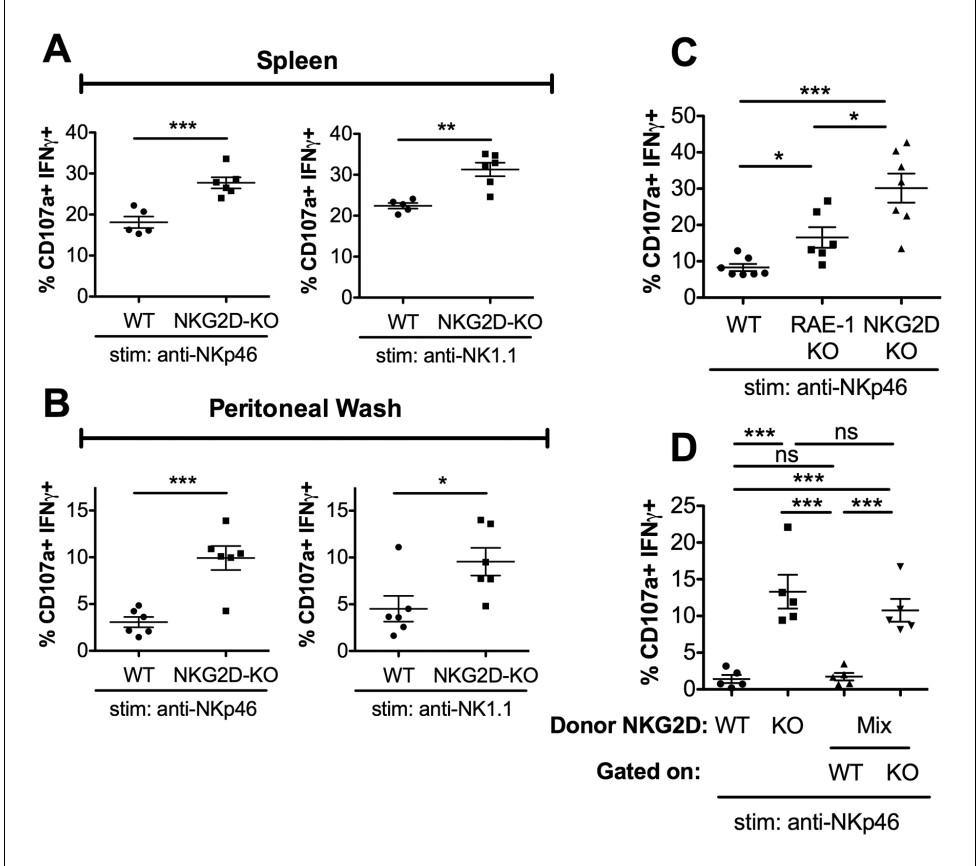

**Figure 3.** RAE-1ε contributes to cell-intrinsic NKG2D-mediated regulation of NK responsiveness. (**A and B**) Percentage of activated splenic or peritoneal NK cells from WT or NKG2D-KO mice after ex vivo stimulation. Data are representative of >4 independent experiments. (**C**) NK activation in WT, RAE-1-KO, and NKG2D-KO peritoneal cells after ex vivo stimulation. Data are representative of three independent experiments. (**D**) NK activation after ex vivo stimulation of peritoneal cells from WT mice 8 weeks after lethal irradiation (11 Gy rad split dose) and reconstitution with bone marrow cells from WT (CD45.1) or NKG2D-KO (CD45.2) mice or a 1:1 mix. Data are representative of two independent experiments. Statistical significance was determined using one-way ANOVA with Bonferroni post-tests (**C, D**) or two-tailed unpaired Student's t tests (**A, B**). Data represent means ± SEM.
DOI: https://doi.org/10.7554/eLife.30881.008

The following figure supplement is available for figure 3:

**Figure supplement 1.** NK cell activation with NKG2D stimulation, and composition of NK cell population in mixed NKG2D bone marrow chimeras.
DOI: https://doi.org/10.7554/eLife.30881.009

We considered that NKG2D-mediated desensitization could happen in a cell-intrinsic manner – that is, through a given NK cell's interaction with ligand and consequent desensitization – or cell-extrinsically via a specific population of 'suppressor' cells. To discriminate between these hypotheses, we generated bone marrow chimeras containing NKG2D-WT and NKG2D-KO cells in the same animal, or singly reconstituted chimeras as controls. WT (CD45.1) mice were lethally irradiated and reconstituted with bone marrow cells from WT (CD45.1) mice, NKG2D-KO (CD45.2) mice, or a 1:1 mixture of the two genotypes. Reconstitution efficiency was consistently greater than 99%, and the mixed chimeric mice contained similar numbers of WT and NKG2D-KO NK cells (*Figure 3—figure supplement 1B*). We then tested the chimeras for NK cell responsiveness. Consistent with our earlier data, NK cells from mice reconstituted with NKG2D-KO bone marrow showed greater responses than NK cells from mice reconstituted with WT bone marrow. Interestingly, NK cells from the mixed chimeras recapitulated these responses, as NKG2D-KO NK cells were hyper-responsive compared with WT NK cells in the same animals (*Figure 3D*). These data demonstrated that NKG2D desensitizes NK responses in a cell-intrinsic manner.

## Endothelial cells in lymph nodes as the primary source of endogenous RAE-1ε

We next sought to identify the cellular source of RAE-1ε responsible for engaging NKG2D and desensitizing NK cells. We used a bone marrow chimera approach to restrict RAE-1ε expression to hematopoietic or nonhematopoietic cells. We used a radiation dose (600 Gy + 500 Gy split dose) that reliably led to replacement of >99% of cells in the hematopoietic compartment, although we cannot exclude the presence of some radio-resistant bone-marrow-derived cells in the chimeras. After irradiation, WT or RAE-1-KO mice were reconstituted with bone marrow from WT or RAE-1-KO mice, and NKG2D cell surface levels were analyzed on NK cells 8 weeks after reconstitution. As expected, KO → KO chimeras showed substantially higher NKG2D levels compared with WT → WT controls (Figure 4A) (Figure 4-figure supplement 1A). Chimeric mice in which RAE-1ε was present only in hematopoietic cells (WT → KO) showed high NKG2D levels comparable to KO → KO chimeras, indicating that hematopoietic RAE-1ε does not play a major role in engaging NKG2D, although there was a reproducibly small effect in most experiments that failed to reach significance. In contrast, mice with RAE-1ε expression restricted to nonhematopoietic cells (KO → WT) completely recapitulated the low NKG2D levels seen in WT → WT animals (*Figure 4A*) (*Figure 4—figure supplement 1A*). When we analyzed the functional responses of NK cells in these chimeras, a similar pattern emerged, with nonhematopoietic RAE-1 playing a dominant role in the desensitization of NK responses, although hematopoietic RAE-1 did show some effect (*Figure 4B*). These data suggested that nonhematopoietic cells are the dominant source of RAE-1ε that engages NKG2D and regulates NK cell responsiveness.

We then began a search for the nonhematopoietic source of RAE-1ε. Because RAE-1-KO mice had elevated NKG2D levels on NK cells in blood and other peripheral tissues, we reasoned that the cellular source of RAE-1ε must be accessible to these NK cells as part of their normal circulatory pattern. Therefore, we used flow cytometry to analyze RAE-1ε on nonhematopoietic cells in various organs encountered by circulating NK cells. Like other lymphocytes, circulating NK cells navigate to and from blood and secondary lymphoid organs. Lymph nodes are central hubs for circulating lymphocytes and have crucial regulatory roles. After gentle enzymatic dissociation of lymph nodes, four populations of nonhematopoietic (CD45-neg) lymph node cells can be delineated by expression of the adhesion molecule CD31 and the transmembrane protein Podoplanin (PDPN) (*Figure 4—figure supplement 1B*) (*Turley et al., 2010*). Cells that are CD31+ PDPN-neg are blood endothelial cells (BECs) and CD31+ PDPN+ cells are lymphatic endothelial cells (LECs). Lymphocytes intimately engage these endothelial cells to enter and exit lymph nodes (*Butcher et al., 1986*). CD31-neg PDPN+ cells are fibroblastic reticular cells (FRCs), which comprise a flexible cellular matrix that defines the lymph node architecture (*Turley et al., 2010*). The CD31-neg PDPN-neg double negative (DN) population is poorly characterized.

We isolated inguinal lymph nodes from naive B6 mice and used flow cytometry to analyze RAE-1ε on these four populations. Whereas DN cells and FRCs showed little to no RAE-1ε, we found substantial RAE-1ε expression on BECs and LECs (*Figure 4C*). This was not due to promiscuous binding of the RAE-1ε antibody, because the staining completely disappeared in RAE-1-KO mice (*Figure 4—figure supplement 1C*). Next, we examined whether RAE-1ε was expressed on endothelial cells in other tissues. Splenic CD31-hi endothelial cells did express low amounts of RAE-1ε, but endothelial cells in the lung, liver, and heart showed little to no RAE-1ε (*Figure 4D* and *Figure 4—figure supplement 2*); all other nonhematopoietic cells in these cell preparations were also negative for RAE-1ε (not shown).

High Endothelial Venule (HEV) endothelial cells are a specialized subset of BECs that mediate lymphocyte entrance into lymph nodes (*Berg et al., 1989*). HEV cells can be identified using the antibody MECA-79, which recognizes a specific carbohydrate motif (*Figure 4—figure supplement 1D*) (*Streeter et al., 1988*). Interestingly, RAE-1ε expression was substantially higher on HEV cells than the average expression on non-HEV BECs (*Figure 4—figure supplement 1E*).

In summary, these experiments showed that nonhematopoietic cells are the dominant compartment responsible for steady state RAE-1ε-mediated NKG2D engagement and NK desensitization, and our analysis of cellular RAE-1ε expression implicate endothelial cells in secondary lymphoid tissue as the relevant cellular source for RAE-1ε. These findings suggest a model in which NK cells,

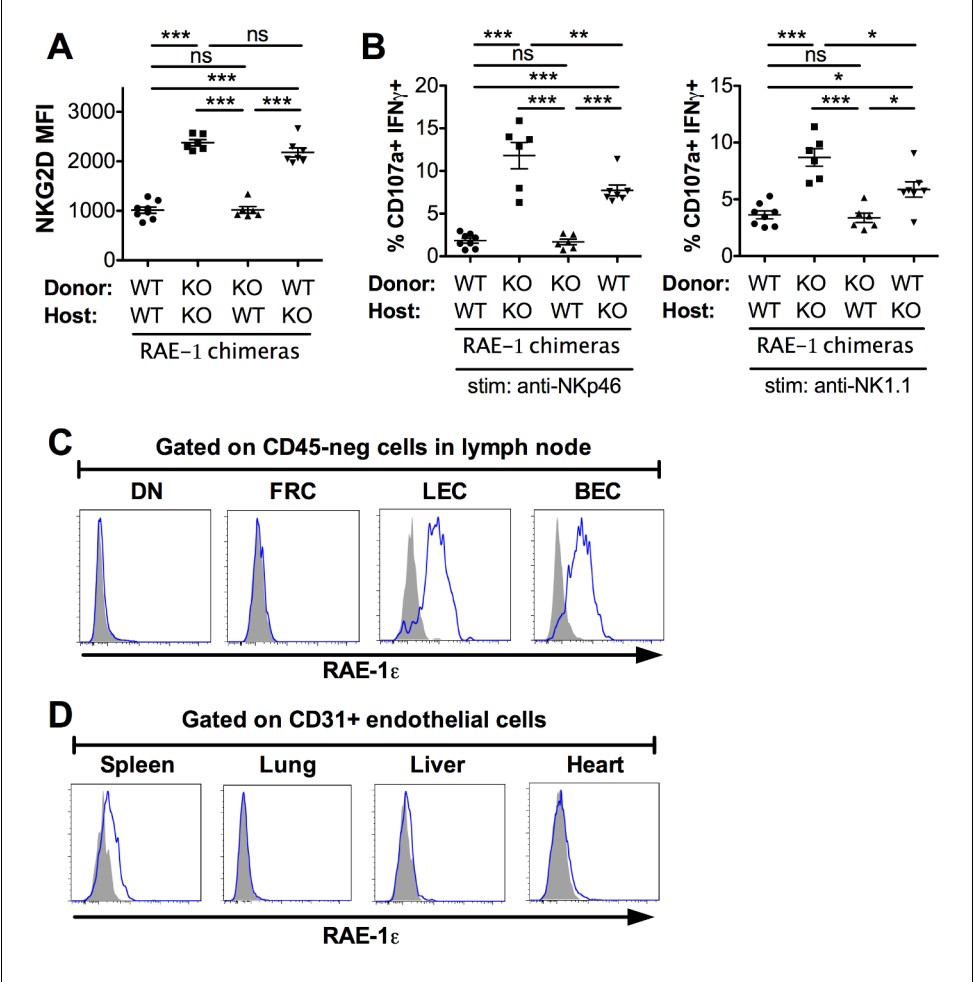

**Figure 4.** Lymph node endothelial cells as the endogenous source of RAE-1ε. (A) NKG2D cell surface levels on blood NK cells 8 weeks after WT or RAE-1-KO mice were lethally irradiated and reconstituted with WT or RAE-1-KO bone marrow. Data are representative of three independent experiments. (B) Percentage of activated NK cells from peritoneal cells from WT and RAE-1-KO bone marrow chimeras after plate-bound antibody stimulation. Data are representative of two independent experiments. (C) RAE-1ε expression on the indicated CD45-neg stromal cell populations in inguinal lymph nodes from WT mice. Data are representative of >4 independent experiments. (D) RAE-1ε expression gated on CD45-neg; Ter119-neg; CD31+ endothelial cells in the indicated organs. Data are representative of three independent experiments. Statistical significance was determined using one-way ANOVA and Bonferroni post-tests. Data represent means ± SEM.

DOI: https://doi.org/10.7554/eLife.30881.010

The following figure supplements are available for figure 4:

**Figure supplement 1.** Spleen and peritoneal wash NKG2D expression in RAE-1 bone marrow chimeras, and expression of RAE-1 on endothelial cells and high endothelial venules.

DOI: https://doi.org/10.7554/eLife.30881.011

**Figure supplement 2.** Comparison of RAE-1 expression by endothelial cells in different organs and sites.

DOI: https://doi.org/10.7554/eLife.30881.012

---

trafficking in and out of lymphoid tissue during homeostatic circulation, are continuously engaged and desensitized by RAE-1ε expressed on endothelial cells.

## Endothelial RAE-1ε and NKG2D engagement in the tumor microenvironment

NK cell responsiveness is controlled by systemic interactions and at local sites of inflammation such as the tumor microenvironment (*Joncker et al., 2010*; *Ardolino et al., 2014*). The powerful antitumor activity of NK cells often selects for tumor cells and microenvironments that can circumvent the

NK response (*Marcus et al., 2014*). To study the effects of endogenous RAE-1ε in tumor microenvironments, we used the B16-BL6 (hereafter called B16) model of melanoma, a classic syngeneic tumor model that is sensitive to NK killing but lacks NKG2D ligand expression (*Lakshmikanth et al., 2009*). WT mice were implanted subcutaneously with B16 cells. After establishment of tumors, mice were treated with antibodies against RAE-1δ, RAE-1ε, or both for 48 hr, after which the tumors were harvested, dissociated to single-cell suspensions, and NK cells infiltrating the tumors were analyzed for surface NKG2D levels. Blocking RAE-1ε but not RAE-1δ caused dramatic NKG2D upregulation on tumor-infiltrating NK cells (*Figure 5A*). Because B16 tumors completely lack expression of NKG2D ligands, we suspected an endogenous source of RAE-1ε, so we analyzed NK cells infiltrating B16 tumors implanted in WT or RAE-1-KO mice. Tumor-infiltrating NK cells mice had elevated NKG2D in RAE-1-KO mice compared with WT controls (*Figure 5B*). Similar results were obtained when tumor-infiltrating NK cells were examined in mice implanted subcutaneously with syngeneic RMA-S lymphoma cells, which also completely lack NKG2D ligand expression (*Figure 5B*).

We analyzed RAE-1 bone marrow chimeras to determine the cellular compartment of RAE-1ε in the tumor microenvironment. We found that nonhematopoietic RAE-1ε was dominant in downregulating NKG2D in B16 tumors (*Figure 5C*), although in some experiments hematopoietic RAE-1 molecules seemed to show some variable effect, albeit not statistically significant (*Figure 5—figure supplement 1A*).

We analyzed single-cell suspensions from B16, RMA-S, and TRAMP-C2 (prostate adenocarcinoma model) and found that tumor-associated endothelial cells expressed copious RAE-1ε (*Figure 5D*) in all tumor models tested. The staining specificity was confirmed using the RAE-1-KO mice as genetic controls (*Figure 5—figure supplement 1B*). When we quantified RAE-1ε levels on tumor-associated endothelial cells compared with lymph node BECs, we saw a much greater expression of RAE-1ε on tumor-associated ECs (*Figure 5E*), indicating that the tumor microenvironment is a substantial inducer of endothelial RAE-1ε.

To address whether these findings applied to tumors that arise naturally, we explored endothelial RAE-1ε expression in the genetically engineered 'KP' cancer model (*DuPage et al., 2012*). KP mice contain germline mutations targeting loxP sites to the *Trp53* tumor suppressor gene and a lox-P-flanked STOP cassette preceding an oncogenic *Kras^{G12D}* allele. Viral delivery of Cre recombinase results in deletion of p53 and expression of oncogenic KRAS in vivo, leading to tumorigenesis in the injected tissue (*DuPage et al., 2009*). We injected lentivirus expressing Cre into the hind leg muscle of KP mice to generate autochthonous sarcomas. Established KP sarcomas or matched healthy tissue from the opposite leg were dissociated and analyzed for endothelial RAE-1ε. Healthy leg muscle lacked endothelial RAE-1ε expression, whereas endothelial cells in KP sarcomas showed robust RAE-1ε induction (*Figure 5F*).

Together, these data suggested that: (1) tumor-infiltrating NK cells are engaged by non-tumor RAE-1ε; (2) nonhematopoietic cells are the primary endogenous source of RAE-1ε responsible for NKG2D engagement; and (3) endothelial cells in transplanted tumors and autochthonous models of mouse cancer are induced to express especially high amounts of RAE-1ε. These data are consistent with a model in which NK cells recruited to tumors are engaged by RAE-1ε induced on endothelial cells in the tumor microenvironment.

## Endogenous RAE-1ε - NKG2D interactions mitigate NK responses to tumors in vivo

NK cells protect the host from tumors in vivo, and the strength of the NK antitumor response depends on the intrinsic responsiveness of NK cells (*Ardolino et al., 2014*). Therefore, we hypothesized that disrupting interactions between NKG2D and host RAE-1ε would amplify the antitumor NK response in vivo. However, the situation is complicated by the fact that NKG2D ligands are often present on tumor cells, and this interaction provides an acute activation signal that kills tumor cells and protects the host. We undertook a series of experiments to clarify the effect of NKG2D ligands on host cells vs. tumor cells.

First, we turned to the B16 model, which is sensitive to NK cell killing but does not express NKG2D ligands. B16 tumors can be injected intravenously (metastasis model) or subcutaneously (solid tumor model). We reasoned that NKG2D-KO mice should show enhanced protection from B16 tumors given the observed NK hyper-responsiveness in these mice. WT and NKG2D-KO mice were injected intravenously with a limiting number of B16 cells and monitored for survival.

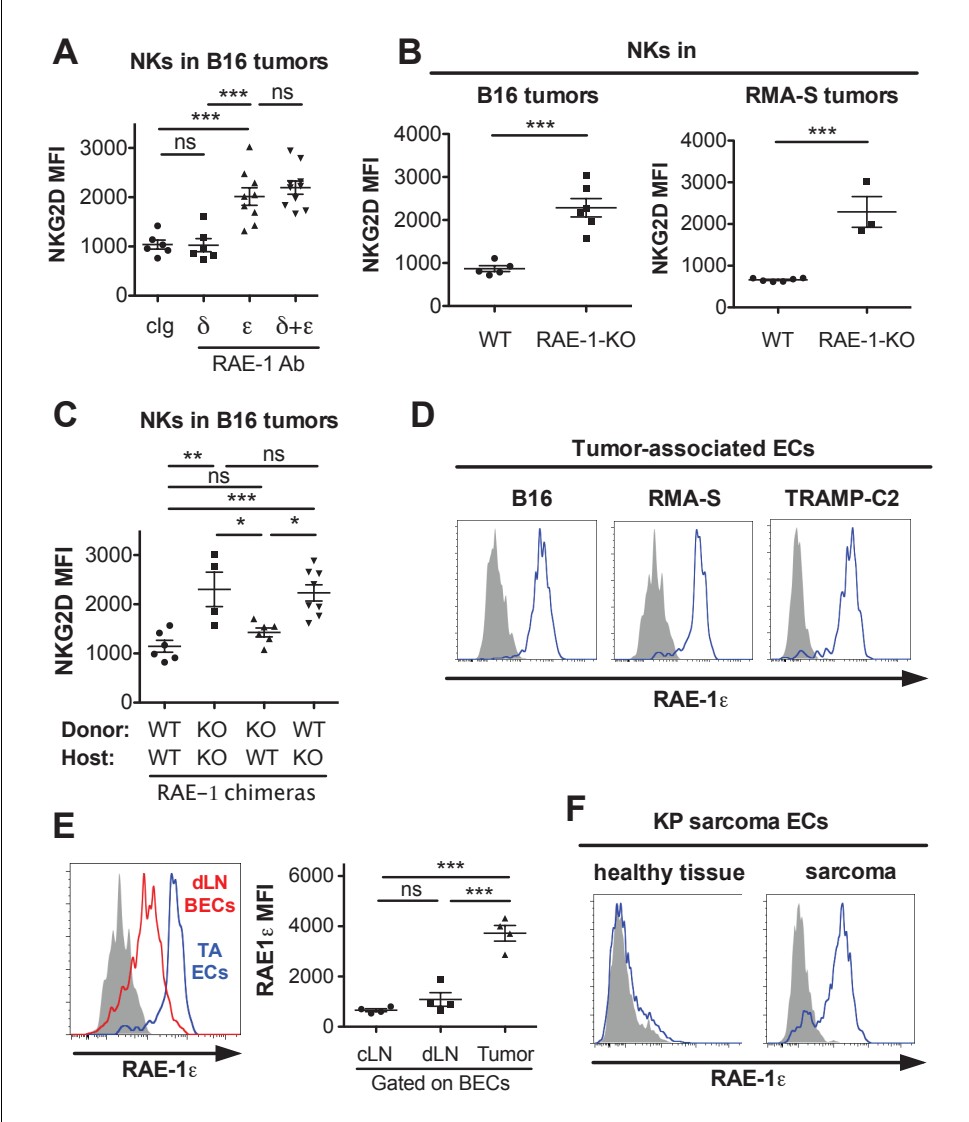

**Figure 5.** Endothelial RAE-1ε and NKG2D engagement in the tumor microenvironment. (**A**) NKG2D surface levels on NK cells infiltrating subcutaneous B16 tumors in mice 48 hr after injection of the indicated antibody. Data are representative of three independent experiments. (**B**) NKG2D surface levels on NK cells from dissociated B16 or RMA-S subcutaneous tumors in WT or RAE-1-KO mice. Data are representative of >4 independent experiments. (**C**) NKG2D surface levels on NK cells infiltrating established B16 tumors in WT or RAE-1-KO chimeric mice. Data are representative of three independent experiments. (**D**) RAE-1ε expression gated on tumor-associated endothelial cells in WT mice with established B16, RMA-S, or TRAMP-C2 tumors. Data are representative of >4 independent experiments. (**E**) RAE-1ε expression on tumor-associated endothelial cells (TA ECs) or blood endothelial cells in draining inguinal lymph nodes (dLN BECs) or contralateral control lymph nodes (cLN BECs) from WT mice with established B16 tumors. Data are representative of >4 independent experiments. (**F**) RAE-1ε expression on endothelial cells in hind leg sarcomas from KP mice or matched healthy tissue from the other hind leg. Data are representative of three independent experiments. Statistical significance was determined using one-way ANOVA and Bonferroni post-tests (**A, C, E**) or a two-tailed unpaired Student's t test (**B**). Data represent means ± SEM.

DOI: https://doi.org/10.7554/eLife.30881.013

The following figure supplement is available for figure 5:

**Figure supplement 1.** Analysis of tumor-associated NKG2D levels in RAE-1 chimeras and endothelial RAE-1ε staining in RAE-1-KO mice.

DOI: https://doi.org/10.7554/eLife.30881.014

Consistent with our hypothesis, NKG2D-KO mice showed reduced and delayed mortality compared with matched WT controls (*Figure 6A*). Importantly, NK depletion resulted in dramatically accelerated mortality in WT and NKG2D-KO mice, eliminating the protective effect of NKG2D deficiency. When WT or NKG2D-KO mice were implanted with B16 cells subcutaneously, NKG2D-KO mice were

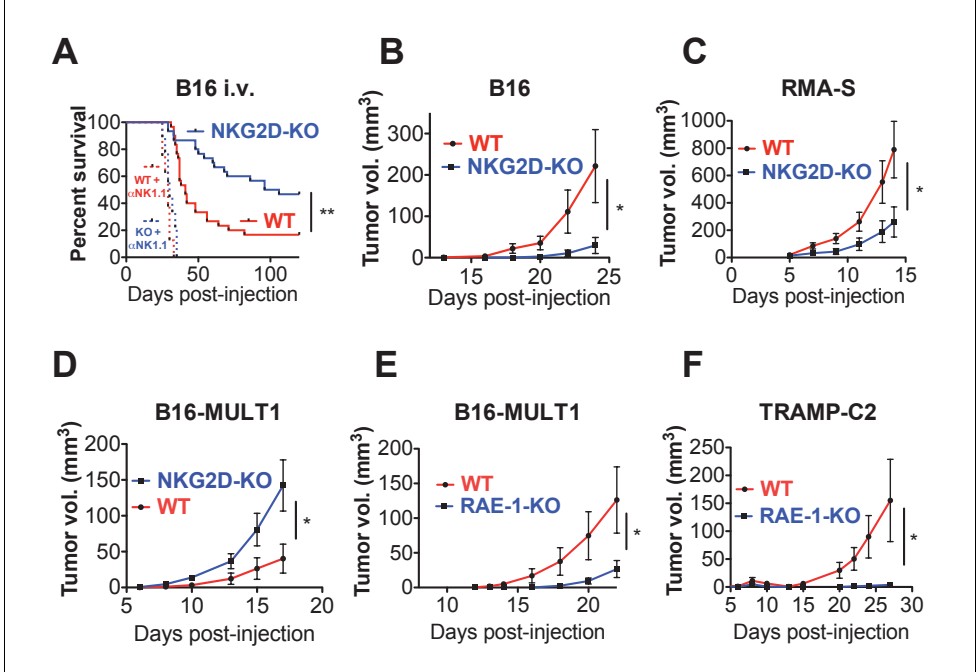

**Figure 6.** Endogenous RAE-1ε - NKG2D interactions limit NK responses to tumors. (**A**) WT or NKG2D-KO mice (n = 29–30) were challenged with 2 × 10^4 B16 cells i.v. and monitored for morbidity. Matched groups of each strain (n = 9–10) were depleted of NK cells before implanting the tumor cells. Data are combined results of three independent experiments. (**B**) WT or NKG2D-KO mice (n = 9) were challenged with 5 × 10^3 B16 cells s.c. and monitored for tumor growth. Data are representative of three independent experiments. (**C**) WT or NKG2D-KO mice (n = 9) were challenged with 5 × 10^5 RMA-S cells s.c. and monitored for tumor growth. Data are representative of three independent experiments. (**D**) WT or NKG2D-KO mice (n = 9) were challenged with 5 × 10^4 B16-MULT1 cells s.c. and monitored for tumor growth. Data are representative of two independent experiments. (**E**) WT or RAE-1-KO mice were challenged with 5 × 10^4 B16-MULT1 cells s.c. and monitored for tumor growth. (n = 12) Data are representative of three independent experiments. (**F**) WT or RAE-1-KO mice (n = 7–8) were challenged with 2 × 10^6 TRAMP-C2 cells s.c. and monitored for tumor growth. Data are representative of three independent experiments. Statistical significance was determined using two-way ANOVA. Data represent means ± SEM.

DOI: https://doi.org/10.7554/eLife.30881.015

The following figure supplement is available for figure 6:

**Figure supplement 1.** Endogenous RAE-1-NKG2D interactions limit anti-tumor responses in mice lacking T and B cells.

DOI: https://doi.org/10.7554/eLife.30881.016

more capable of controlling B16 tumor growth than WT counterparts (*Figure 6B*). We also found that Rag2/NKG2D double knockout mice resisted B16 tumors better than Rag2-KO mice, indicating that T cells (and B cells) are not required for the protective effect of NKG2D deficiency (*Figure 6—figure supplement 1A*). A subsequent experiment on the Rag2-KO background showed that treatment of NKG2D-WT mice with anti-NKG2D antibody in vivo starting the day before tumor cell implantation resulted in enhanced control of B16 tumors comparable to the NKG2D-KO mice (*Figure 6—figure supplement 1B*). We then turned to the RMA-S lymphoma transplant tumor model, which is targeted in vivo by NK cells through 'missing self' recognition but also does not express NKG2D ligands. NKG2D-KO mice showed better control of subcutaneous RMA-S tumors than did their WT counterparts (*Figure 6C*). These data indicate that NKG2D expression mitigates in vivo NK responses to tumors that lack NKG2D ligands, and that this desensitizing effect is reversed by acute antibody blockade of NKG2D.

Because most tumors express NKG2D ligands, we analyzed tumor growth of B16 tumor cells transduced to express high surface levels of the NKG2D ligand MULT1. MULT1 is a high-affinity NKG2D ligand that stimulates a strong acute NK cell response that enhances tumor rejection in WT mice (*Figure 6—figure supplement 1C*). Despite the NK hyper-responsiveness in NKG2D-deficient animals, NKG2D-KO NK cells cannot recognize MULT1, and the growth of B16-MULT1 tumors was accelerated in NKG2D-KO mice compared with WT controls (*Figure 6D*). These data are consistent

with previous studies showing a strong protective role for NKG2D interactions with tumor NKG2D ligands (*Cerwenka et al., 2001*; *Diefenbach et al., 2001*; *Guerra et al., 2008*).

Because we had found that RAE-1-KO mice have elevated NKG2D surface levels and enhanced NK responses, we hypothesized that RAE-1-KO mice would show enhanced protection from NKG2D ligand-positive tumors. Indeed, RAE-1-KO mice challenged with B16-MULT1 tumors exhibited superior tumor rejection compared with WT mice (*Figure 6E*). We then turned to a transplant tumor model with endogenous NKG2D ligand expression, the TRAMP-C2 model of prostate adenocarcinoma. TRAMP-C2 cells express RAE-1δ and RAE-1ε and are sensitive to NK cells, and this sensitivity is partly dependent on NKG2D (*Jamieson et al., 2002*). When injected subcutaneously into WT and RAE-1-KO mice, the RAE-1-KO mice showed enhanced control of TRAMP-C2 tumors compared with WT controls (*Figure 6F*).

Collectively, these data demonstrate that interactions between NKG2D and endogenous RAE-1ε desensitize NK responses to tumors in vivo, whereas NKG2D ligands expressed on tumor cells promote antitumor NK responses.

## Discussion

The studies in this paper add considerably to our understanding of the receptors and ligands that regulate NK activity at steady state and in cancer. We show that the activating receptor NKG2D is engaged by endogenous RAE-1ε in healthy WT mice, causing constitutive downregulation of NKG2D from the NK cell surface, and leads to an intrinsic global desensitization of NK cells to acute stimulation. These effects were evidenced by an increase in the responsiveness of NK cells from NKG2D-KO or RAE-1-KO mice to stimulation through diverse activating receptors. Antibody blockade of RAE-1ε in normal mice also resulted in elevated NKG2D levels and increased NK cell responsiveness (*Figure 1A,B*). Moreover, preventing NKG2D interactions with host RAE-1ε enhanced NK cell antitumor responses in vivo. Bone marrow chimera experiments identified nonhematopoietic cells as the dominant source of RAE-1ε, and endothelial cells were found to express RAE-1ε constitutively in lymph nodes. In tumors, RAE-1ε expression on tumor-associated endothelial cells was superinduced compared to endothelial cells in lymph nodes. These data support a model in which endothelial RAE-1ε interacts with NKG2D to desensitize NK cells and diminish antitumor NK responses.

NK cell hyper-responsiveness in NKG2D-KO mice has been reported previously by several groups including ours (*Zafirova et al., 2009*; *Sheppard et al., 2013*; *Deng et al., 2015*), although the trend of increased responsiveness did not reach statistical significance in our original characterization of the NKG2D-KO mouse (*Guerra et al., 2008*). The data presented in this study build on those findings by showing that NKG2D engagement of endogenous ligands desensitizes NK cells, reducing antitumor responses. At the same time, our results support the classical view that NKG2D can participate in immune surveillance of cancer by recognizing NKG2D ligands on tumor cells to induce NK cell activation and tumor cell killing (*Diefenbach et al., 2000*; *Cerwenka et al., 2001*; *Diefenbach et al., 2001*; *Jamieson et al., 2002*; *Guerra et al., 2008*). Hence, NKG2D confers opposing effects, both promoting killing of tumor cells that express NKG2D ligands, and desensitizing NK cells through interactions with host ligands.

We propose that the net outcome of these opposing effects of NKG2D depends on the complement of NK-activating ligands that a given tumor cell expresses, among other factors. For tumor cells that express abundant NKG2D ligands, the ability of NKG2D to promote acute activation against the tumor can outweigh the opposing desensitizing interactions with host cells. In this circumstance, loss of NKG2D results in reduced tumor killing, as exemplified by the observation that NKG2D-deficient mice show reduced killing/rejection of B16-MULT1 tumors and YAC-1 cells (*Figure 6D*, *Figure 2—figure supplement 1B*) (*Jamieson et al., 2002*). We speculate that this situation also applies to previous findings that NKG2D-KO mice were defective in immune surveillance of spontaneous tumors that express NKG2D ligands, such as prostate adenocarcinomas in TRAMP mice and lymphomas in Eu-Myc mice (*Guerra et al., 2008*). In contrast, for tumor cells that express abundant other NK-activating ligands, and either do not express NKG2D ligands (e.g. RMA-S and B16 cells), or depend little on NKG2D ligands for NK killing, the global desensitization of NK cells caused by NKG2D interactions with host ligands is predicted to exert a dominant effect. In this situation, NKG2D deficiency results in enhanced tumor killing (*Figure 6B,C*). Interestingly, a recent study reported that in a spontaneous model of hepatocellular carcinoma, NKG2D-KO mice exhibited a

reduced tumor incidence (*Sheppard et al., 2017*). We speculate that NKG2D-mediated desensitization may be dominant in that model, though the authors offered an alternative explanation for their findings. It is also possible that some tumor environments contain signals that reverse NKG2D-mediated desensitization while preserving NKG2D-mediated activation, or vice versa. Thus, the net effect of NKG2D will likely depend on the activating and inhibitory ligands tumor cells express, as well as the context of the tumor microenvironment.

Although endogenous RAE-1ε played a significant role in NK desensitization, it does not completely account for the desensitizing activity of NKG2D, based on the finding that NK cells attained higher responsiveness in NKG2D-KO mice than in RAE-1-KO mice (*Figure 3C*). While our data clearly support a model in which endogenous interactions of NKG2D with RAE-1ε cause steady-state NK cell desensitization, we do not rule out other potential mechanisms contributing to hyper-responsiveness of NKG2D-KO cells – such as interactions of NKG2D with other ligands at steady state, or 'tonic' signaling through NKG2D, at steady state or during NK development.

Perhaps the most surprising finding in this report is the expression and functional relevance of RAE-1ε on endothelial cells. NKG2D ligand expression in adult mice has been thought largely restricted to cells undergoing certain stress responses associated with oncogenesis or infection, but here we show that blood endothelial cells and lymphoid endothelial cells in secondary lymphoid organs of healthy animals express RAE-1ε at steady-state. It was striking that, in the same mice, endothelium in several other tissues was completely negative for RAE-1ε. These findings suggest that the lymphoid tissue environment imparts specific signals that upregulate RAE-1ε on associated endothelial cells. Specific expression on endothelial cells in secondary lymphoid tissue further suggests that the system may be designed to desensitize NK cells that traverse this tissue. It is important to note that our data clearly identify non-hematopoietic cells as the dominant source of RAE-1ε-mediated NK desensitization, but we cannot conclusively establish endothelial cells as relevant desensitizing compartment without an endothelial-specific RAE-1ε-KO mouse. We were also surprised to find that resident liver NK cells (CD49a-neg CD49b+) were engaged by host RAE-1ε. We do not know the cellular source of RAE-1ε responsible for engaging these NK cells, as we detected little to no RAE-1ε in cell suspensions from liver. We speculate that resident liver NK cells might traffic to distinct anatomical structures, perhaps liver-specific lymphoid tissue, that might contain RAE-1ε-expressing cells.

We found especially high levels of RAE-1ε on endothelial cells in the tumor microenvironment. Notably, RAE-1ε was expressed on vasculature in all tumor models tested, including transplant models of melanoma, lymphoma, prostate cancer, and the genetically engineered KP model of autochthonous sarcoma. In all cases, levels of tumor-associated endothelial RAE-1ε were even greater than on endothelial cells in draining and non-draining lymph nodes (*Figure 5E*). We can only speculate as to the mechanism of RAE-1ε induction on these cells. Vasculature in the tumor microenvironment is extremely dynamic, characterized by extensive angiogenesis. RAE-1 expression has been linked to high levels of cellular proliferation in development and wound healing (*Jung et al., 2012*). Perhaps a similar mechanism is responsible for the super-induced expression of RAE-1ε on neovasculature in tumors. Steady-state RAE-1ε expression on lymph node endothelial cells, but not endothelium in normal non-lymphoid tissues, is more surprising. Clearly, more work is required to fully understand NKG2D ligand regulation in these cells.

Endothelial cells are sometimes thought of as passive circulatory conduits. Our data and other studies clearly paint a more complex picture. Endothelial cells have recently been shown to produce immunomodulatory cytokines, and lymph node endothelial cells can tolerize peripheral T cells by presenting self MHC – peptide complexes (*Rouhani et al., 2015*). The data in our study add to the emerging role of endothelial cells as active regulators in the immune response by suggesting that endothelial cells regulate NK cell activity. In a way, this seems logical, given the inevitable and intimate contact a lymphocyte must make with the vasculature upon entering and exiting lymph nodes and other organs during homeostatic circulation or inflammation. Because multiple NK cell receptors are known to regulate NK cell responsiveness, we are intrigued by the possibility that endothelial cells may also regulate NK cell activity through other receptor-ligand systems. In addition, it is interesting to speculate whether NK cells have effects on the endothelial cell biology, through NKG2D-RAE-1ε or other mechanisms.

We wonder if endothelial-RAE-1ε-mediated NK desensitization is important in pathological contexts other than cancer. It is tempting to speculate that signals from vasculature may help inform

infiltrating NK cells, or other lymphocytes, about the inflammatory state of the tissue. Perhaps NKG2D ligand expression on endothelial cells is advantageous for preventing autoreactivity and/or limiting inflammatory signals during the resolution of infection. Perhaps, tumor microenvironments mimic other physiological states, such as wound healing, for which RAE-1ε induction and NK desensitization is beneficial to the host. Endothelial RAE-1ε clearly regulates NK responses to tumors, but the relevant contribution of lymph node vs. tumor endothelium remains unclear. It is plausible that both environments control NK desensitization. We also speculate that physiological situations may arise in which NK cells transit from a desensitizing environment to one that reverses desensitization. Our data show that at steady state, NKG2D internalization is almost completely reversed as early as 12 hr after antibody blockade of RAE-1ε, and that NK function can be elevated at 48 hr after blockade. Further study is needed to understand how physiological changes in the host's desensitizing environment might regulate the kinetics of the NK response.

The differences in intracellular signaling mechanisms that culminate in NKG2D-mediated acute NK activation versus desensitization remain poorly understood. An attractive hypothesis is that receptor engagement in the absence of inflammatory signals results in initial activation followed by desensitization; if inflammatory cytokines are present, as in an infection, the activation response might be sustained. This model fits with evidence that inflammatory cytokines – such as IL-12/1 L-18 or IL-2/15 family cytokines – sustain or reinvigorate desensitized NK responses (*Ardolino et al., 2014*). Similarly, activation vs. desensitization may be modulated by cell surface signals on the surface of the target cell. It might also be considered that different qualities of the encounter with ligand could result in divergent outcomes. For example, differences in the duration and/or frequency of the receptor-ligand interactions could contribute to opposing outcomes, or perhaps the relative affinity of a given ligand for its receptor, coupled with amount of ligand expressed, could potentially affect the signaling outcome. These and other molecular mechanisms might determine why interactions with some cells but not others results in desensitization.

It is notable that a previous study did find a role for desensitization of NK cells by tumors lacking MHC I (*Ardolino et al., 2014*), and tumor cells expressing NKG2D ligands can cause global NK cell desensitization in vitro (*Coudert et al., 2008*). Furthermore, transgenic over-expression of NKG2D ligands is associated with NK cell desensitization as well (*Oppenheim et al., 2005*). Thus, endothelial cells may not have unique desensitizing activity per se but rather could have an important role in NK desensitization by virtue of expressing NKG2D ligands (which most healthy cells lack) and/or their unique physiological role in extravasation, intravasation, and circulation. Furthermore, it is possible that the impact of endothelial cell RAE-1 on tumor rejection is to desensitize NK cells before they encounter tumor cells, whereas NKG2D ligands on tumor cells might affect NK cells differently after extravasation; these dynamics may also depend on the inflammatory state of the tumor microenvironment. Clearly, more work is needed to better understand the mechanisms that regulate NK cell activation and desensitization and its relation to tumor rejection.

Our data may have translational implications. We hypothesize that treatments to disrupt interactions between NK cells and NKG2D ligands on vasculature, while preserving interactions with NKG2D ligands on tumor cells, may have powerful therapeutic benefits. This could be accomplished by antibody blockade of ligands expressed by endothelial cells but not tumors. Understanding the mechanisms supporting endothelial NKG2D ligand expression could also reveal targets for specific pharmacological inhibition.

## Materials and methods

### Mice and in vivo procedures

C57BL/6J mice were bred from mice obtained from The Jackson Laboratory (Bar Harbor, ME). NKG2D-KO mice were previously generated in our lab as described (*Guerra et al., 2008*). RAE-1-KO mice were previously generated in our lab using CRISPR-Cas9 and guide RNAs targeting the open-reading frames of the *Raet1d* and *Raet1e* genes, as described (*Deng et al., 2015*). KP mice contain inducible mutations in the proto-oncogene *Kras* and the tumor suppresser gene *Trp53* and were bred from mice obtained from The Jackson Laboratory. All mice were maintained at the University of California, Berkeley in accordance with guidelines from the Animal Care and Use Committee. Sex- and age-matched (8- to 12-week-old) mice were used for the experiments. In most

experiments, KO mice were compared with cousin WT controls (i.e. derived from the same grand-parents) and were co-housed to minimize confounding genetic and environmental variables.

In vivo blockade of NKG2D ligands was achieved by peritoneal injection of 100 µg of the indicated antibody. Antibodies against RAE-1δ (clone 199205) and RAE-1ε (clone 205001) were obtained from R&D Systems (Bio-Techne Corp, Minneapolis, MN). Anti-MULT1 (clone 1D6) was a kind gift from Stipan Jonjic. We confirmed the blocking efficacy of these reagents before in vivo use (*Figure 1—figure supplement 1A*).

All transplant tumor models were injected i.v. (metastasis model) or s.c. (solid tumor model) by injection with an insulin syringe (BD Biosciences, San Jose, CA). For injection, tumor cells were suspended in 100 µl PBS and injected by the indicated route. In some groups, NK cells were depleted by twice weekly injections of 100 µg anti-NK1.1 antibody (clone PK136) beginning the day before tumor injection. Tumor growth was measured by caliper, and tumor volume was calculated using the modified ellipsoid formula: $V = 0.5 \times [(length + width)/2]^3$.

In some experiments, adult splenocytes were labeled with 1 µg/ml CFSE (Biolegend, San Diego, CA) according to the manufacturers instructions, and $5 \times 10^7$ labeled cells were transferred to recipient mice by i.v. injection.

For bone marrow chimera experiments, recipient mice were lethally irradiated with 11 Gy (6 Gy +5 Gy split dose) using an X-ray irradiator and then given $1 \times 10^7$ donor bone marrow cells by i.v. injection. Mice were allowed to recover and reconstitute for at least 8 weeks before analysis or tumor injection.

Spleens were dissociated by mashing through a 70 µM filter into PBS. To dissociate lymph nodes, lungs, heart, liver, or tumors for flow cytometry, organs were gently dissociated with according to a published protocol optimized for stromal cell analysis (*Broggi et al., 2014*). Briefly, organs were mechanically dissociated using a sharp blade, and then incubated in complete media with 3.5 mg/ml Collagenase D, 1 mg/ml Collagenase IV for 30 min at 37°C with rotation. Cells were then pipetted up and down rigorously 100 times to create a single cell suspension, with additional rounds of 10-min incubation followed by pipetting as needed.

## RNA, cDNA, and qPCR

Total RNA was isolated from cells using the RNeasy kit (Qiagen, Hilden, Germany) and converted to cDNA using the iScript system (Bio-Rad, Hercules, CA) according to the manufacturer's instructions. cDNA was subjected to real-time PCR using SsoFast EvaGreen supermix (Bio-Rad) in the presence of primers to amplify *Klrk1* mRNA, or the transcripts of the housekeeping genes β-actin and Rpl19, in a CFX96 RT-qPCR thermocycler (BioRad). Relative mRNA values for *Klrk1* were normalized to the levels of the housekeeping genes, using CFX96 software.

## Cell culture

All cell cultures were performed in a humidified 37°C incubator at 5% $CO_2$. Cells were cultured in DMEM or RPMI media (Life Technologies, Carlsbad, CA) supplemented with 5% fetal calf serum (Omega Scientific, Tarzana, CA), 0.2 mg/ml glutamine, 100 U/ml penicillin, 100 µg/ml streptomycin (Sigma–Aldrich, St. Louis, MO), 10 µg/ml gentamicin sulfate (Lonza, Basel, Switzerland), and 20 mM HEPES (Thermo Fisher Scientific, Waltham, MA).

## Flow cytometry and FACS

For all flow cytometry experiments, single-cell suspensions were generated and incubated for 20 min with supernatant from the 2.4G2 hybridoma to block FcγRII/III receptors, followed by incubation with fluorochrome- or biotin-conjugated specific antibodies for an additional 20 min. In some experiments, an additional incubation with fluorophore-conjugated streptavidin (Biolegend) was performed. For intracellular staining, cells were fixed and permeabilized using the Cytofix/Cytoperm kit (BD Biosciences) before incubation with intracellular antibodies. Samples were analyzed on a LSR Fortessa or LSR Fortessa X20 (BD Biosciences) and data were analyzed with FlowJo software (Tree Star Inc.). Dead cells were excluded from analysis using DAPI (Biolegend) or Live-Dead fixable dead cell stain kits (Molecular Probes) following the manufacturer's instructions. In some experiments, NK cells were sorted using the Influx Cell Sorter (BD Biosciences).

## Antibodies

We used the following antibodies: from Biolegend; anti-CD3ε (clone 145–2 C11), anti-CD4 (clone GK1.5), anti-CD11b (clone M1/70), anti-CD19 (clone 6D5), anti–IFN-γ (clone XMG1.2), anti-NKp46 (clone 29A1.4), anti–NK1.1 (clone PK136), anti-Ter119 (clone TER-119), anti-CD31 (clone 390), anti-Podoplanin (clone 8.1.1), anti-HEV (clone MECA-79), mouse IgG2b isotype control, and rat IgG2b isotype control; from eBioscience; anti-CD27 (clone 37.51), anti-CD45.1 (clone A20), anti-CD45.2 (clone 104), anti-CD107a (clone 1D4B), anti-NKG2D (clone MI-6); from R and D Systems; mouse NKG2D-Fc fusion protein; from Jackson ImmunoResearch; goat anti-mouse IgG. For flow cytometry analysis of RAE-1ε, we used the EZ-Link-Sulfo-NHS-LC biotin kit (Thermo Fisher) to biotinylate clone 205001 mAb (from R and D Systems), the same clone used for in vivo NKG2D ligand blockade.

## NK responsiveness assay

To analyze the responsiveness of NK cells ex vivo, 96-well high-binding flat-bottom plates (Thermo Fisher) were coated overnight with PBS plus the indicated antibody against NK activating receptors. Plate-bound antibodies were coated at the following concentrations: anti-NK1.1: 50 µg/ml; anti-NKG2D: 5 µg/ml; anti-NKp46: 5 µg/ml. Plates were washed three times with PBS before stimulation. Single-cell suspensions were generated from the indicated tissue and cultured in the coated plates for 5 hr in the presence of Golgi-Stop and Golgi-Plug (1:1000 each) (BD Biosciences), 1000 U/ml human IL-2, and fluorophore-conjugated anti-CD107a (0.5 µg/ml) (Biolegend). After stimulation, cells were stained for extracellular markers to identify NK cells and then subjected to intracellular staining for IFN-γ, followed by flow cytometry analysis.

## In vitro cytotoxicity assay

To analyze NK killing of tumor cells in vitro, NK cells from mice of the indicated genotype were pre-activated by a single i.p. injection of 200 µg high-molecular weight Poly I:C (Invivogen, San Diego, CA); peritoneal wash cells were harvested 2 days later and pooled by genotype. YAC-1 target cells were labeled with 20 uCi $^{51}$Cr, washed three times, and plated in 100 µl medium at $1 \times 10^4$ cells per well in a 96-well round bottom plate. 100 µl of medium alone (spontaneous release), 2% Triton-X-100 (maximum release), or effector cells at the indicated effector:target ratios were added to targets, in quadruplicate. Cells were incubated for 4 hr at 37°C. Cells were then pelleted, and 100 µl of supernatant was analyzed by gamma counting. The spontaneous release was, in all cases,<20% of maximum release. The percent specific $^{51}$Cr release was calculated according to the following formula: % specific lysis = 100 x (experimental release – spontaneous release) / (detergent release – spontaneous release).

## Statistics and sample size

All statistical analyses were conducted using Prism software (Graphpad, La Jolla, CA), as indicated in the figure legends. Statistical significance is indicated as follows: *p<0.05, **p<0.01, ***p<0.001. For most data sets, pilot experiments were performed with a small sample size (usually n = 3) to determine approximate experimental variances and effect magnitudes, and this information was used to determine sample sizes for subsequent experiments.

## Acknowledgements

We are grateful to all members of the Raulet lab, as well as former Raulet lab members Weiwen Deng, Benjamin Gowen and Russell Vance, as well as JP Houchins of R&D Systems, for their helpful feedback on this manuscript. We thank Hector Nolla, Alma Valeros, Kartoosh Heydari for their invaluable help with cell sorting and maintenance of the flow cytometry facility at UC-Berkeley. We are grateful to JP Houchins and R and D systems for providing anti-RAE-1 mAbs used in this study, and we thank Eugene Butcher for helpful discussions on endothelial cells. Research reported in this publication was supported by NIH/NCI grants R01-CA093678 (DHR) and F31CA203262 (TWT), and a research grant from Innate Pharma, SAS. The content is solely the responsibility of the authors and does not necessarily represent the official views of the National Institutes of Health or Innate Pharma.

# Additional information

## Competing interests

David H Raulet: Co-founder of Dragonfly Therapeutics, and serves on the Scientific Advisory Boards of Innate Pharma, Aduro Biotech and Ignite Immmunotherapy, has a financial interest in all four companies and received research support from Innate Pharma, and may benefit from commercialization of the results of this research. The other authors declare that no competing interests exist.

## Funding

| Funder | Grant reference number | Author |
| --- | --- | --- |
| National Cancer Institute | R01 CA093678 | David H Raulet |
| Innate Pharma, SAS | | David H Raulet |
| National Cancer Institute | F31 CA203262 | Thornton W Thompson |

The funders had no role in study design, data collection and interpretation, or the decision to submit the work for publication.

## Author contributions

Thornton W Thompson, Conceptualization, Data curation, Formal analysis, Supervision, Funding acquisition, Validation, Investigation, Visualization, Methodology, Writing—original draft; Alexander Byungsuk Kim, P Jonathan Li, Jiaxi Wang, Benjamin T Jackson, Validation, Investigation, Visualization, Writing—review and editing; Kristen Ting Hui Huang, Validation, Investigation, Writing—review and editing; Lily Zhang, Investigation; David H Raulet, Conceptualization, Resources, Supervision, Funding acquisition, Methodology, Project administration, Writing—review and editing

## Author ORCIDs

Alexander Byungsuk Kim  http://orcid.org/0000-0002-6425-4566
David H Raulet  http://orcid.org/0000-0002-1257-8649

## Ethics

Animal experimentation: This study was performed in strict accordance with the recommendations in the Guide for the Care and Use of Laboratory Animals of the National Institutes of Health. All mice were maintained at the University of California, Berkeley and experiments were conducted in accordance with approved protocols from the Animal Care and Use Committee, under protocol number AUP-2015-10-8058.

## Decision letter and Author response

Decision letter https://doi.org/10.7554/eLife.30881.020
Author response https://doi.org/10.7554/eLife.30881.021

# Additional files

## Supplementary files

• Transparent reporting form
DOI: https://doi.org/10.7554/eLife.30881.017

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
