## [Decision Letter]

Thank you for submitting your article "Endothelial cells express NKG2D ligands and desensitize anti-tumor NK responses" for consideration by *eLife*. Your article has been reviewed by three peer reviewers, and the evaluation has been overseen by a Reviewing Editor and Michel Nussenzweig as the Senior Editor. The following individuals involved in review of your submission have agreed to reveal their identity: Mark Smyth (Reviewer #1); Petter Hoglund (Reviewer #3).

The reviewers have discussed the reviews with one another and the Reviewing Editor has drafted this decision to help you prepare a revised submission.

Summary:

This manuscript describes the NK cell activation receptor NKG2D and its ligand, RAE-1ε, that is expressed on lymph node endothelial cells and is upregulated on endothelium in tumors. The authors present data that NKG2D-RAE-1ε interactions lead to desensitization of NKG2D responses and impaired anti-tumor activity, which sheds important light on how NK cell reactivity is regulated. The work thus opens many new avenues for further research.

Essential revisions:

The reviewers and Reviewing Editor have viewed this manuscript favorably but there are some issues that need attention, specifically:

1) Provide in vitro cytotoxicity assays to verify the degranulation assays as recommended by two reviewers.

2) Expression of NKG2D on NK cells from the BM and liver should be presented.

3) Address reproducibility of BM chimera data.

The complete reviews of each reviewer are given below, to provide the details and context, but please note that the consensus statement above lists the areas that need focus. When submitting the revision, please reply to the comments listed above.

Reviewer #1:

In this manuscript entitled "endothelial cell express NKG2D ligands and desensitize anti-tumor NK responses", Thompson et al. investigated the mechanism of the desensitization of NK cell responses focused on the interaction between activating NKG2D receptor and its ligands. Specifically, they found that RAE-1ε, which is constitutively expressed on lymph node endothelial cells, negatively regulates NK cell activity through down-modulation of NKG2D under steady state. Thus, blockade of RAE-1ε by antibodies or lack of RAE-1ε augmented NK cell responses, leading to better tumor control.

Overall, manuscript is well written, and the experiments are carefully performed to support the authors' concept. The findings in this manuscript provide significant insight into NK cell biology.

Any more data concerning the mechanism of NK cell activation versus desensitization would be welcomed.

Though authors clearly showed in vivo tumor growth data, using various models, one missing experiment is in vitro cytotoxicity data.

Do naïve NK cells from RAE-1ε KO more effectively kill target cells (e.g. YAC-1) than WT NK cells?

Reviewer #2:

This manuscript by Thompson et al. from the Raulet group examines NK cell "desensitization" mediated by NKG2D interactions with its ligands, in particular RAE-1ε. The key conclusions from this work are that NKG2D interaction with its ligands tunes NK cell responsiveness level to stimulation, that RAE-1ε is expressed by lymph node endothelium, and these interactions impact responses at the site of tumors. The manuscript is very well written, but some of the summary/discussion that is included in the Results should be moved to Discussion, and the Introduction should be shortened. Limitations that diminish enthusiasm for the study include discrepancies with previously published data by this group, and the lack of human data to support the translational relevance that the authors claim.

1) Figure 1 shows that blockade of RAE-1ε in vivo increased NKG2D surface expression on NK cells in the blood, spleen and lymph nodes, and that RAE-1ε KO mice have increased NKG2D expression on NK cells form the blood, LN, spleen, and peritoneum. Why was the BM excluded, which is a relevant hematopoietic tissue for NK cells, including recirculation? Similarly, what about liver (both conventional and tissue resident) NK cells?

2) Figure 2. Formal cytotoxicity experiments against tumor targets are missing. While CD107a is a surrogate for degranulation, there are situations where CD107a is not directly correlated with cytotoxicity. Further, the degranulation differences appear modest, and may not result in differences in killing.

3) Figure 3. How do the authors reconcile the data included in Figure 3, with their previous reported (Guerra et al. Immunity 20108) Figure 6, where they saw different results with NKG2D KOs? Perhaps I am missing a key aspect of the functional experiment, but in the prior paper NKG2D-KO NK cells had similar IFN-γ responses as wild type mice to these stimuli? Further, killing appeared decreased (not increased) against C1498 and YAC-1 targets? Similarly, killing of tumor targets should be included in this assay as well.

The BM chimera data looks very clear and is well presented. The expression of RAE-1ε on various cells in C and D look clear, but are there summary data for reproducibility? There appears to be higher signal in the spleen, and not all cells and tissues in a mouse have been examined. Notably, bone marrow is missing. The authors should temper their conclusions that lymph node endothelial cells are the "primary" endogenous source of RAE-1ε, since experimental data is not shown to support this conclusion. For example, clear data from an endothelial cell specific KO of RAE-1ε would better support such a conclusion.

4) Overall, more discussion of how their results differ from their prior work and explanations of which conclusion is best supported would strengthen the manuscript. For example, their prior paper indicated that mice with Myc-driven lymphomas occur faster in NKG2D KO mice compared to WT mice. This data seems inconsistent with the data being presented in this manuscript about desensitization, with NKG2D KOs having superior survival after challenge with B16 melanoma?

5) Data demonstrating this is relevant for human tumors would increase enthusiasm and better support the translational impact of these findings.

Reviewer #3:

The paper by Thompson identifies a particular NKG2D ligand in HEV in normal lymph nodes and in tumor vasculature. This ligand is shown do desensitize NK cell reactivity, not only to NKG2D-mediated stimulation but also to other non-related receptors. The results are very clear and shed new important light on how NK cell reactivity is regulated.

From a basic NK cell education perspective, the data is particularly exciting. One key issue is why NKG2D ligands on endothelial cells, but not on tumor cells, downtune NK cell responsiveness? The authors provide some plausible explanations, such as differences in cytokine milieu and the inflammatory environment, but one alternative possibility that comes to mind, given the pivotal role for MHC class I expression levels in NK cell tuning, is if MHC class I molecules influence the tuning effect of NKG2D ligand in the systems described? Is there a difference in MHC class I expression between HEV in normal LN and in tumors, and between endothelial cells and tumor cells? Have the authors looked at the influence of MHC class I in their system and have they excluded that differences in MHC class I expression levels influence RAE-1ε-mediated tuning in normal HEV, tumor cells and HEV in tumor tissue?

On the same note, is function tuned differentially by RAE-1ε in NK cells carrying self Ly49 receptors versus NK cells expressing non-self-inhibitory receptors?

Another question is in which way tuning by RAE-1ε is dynamically regulated? The authors show that NKG2D is upregulated 48 hours after adoptive transfer, a timepoint that fits well with the author's previous notion on retuning by MHC class I in vivo, but a notion of such a long time for retuning has always puzzled me. In order for a retuning effect to be of some use, in my mind, it would need to occur much more rapidly. Our earlier data from DL6 mice suggested a very rapid reversibility of NK cell tolerance (4 hours – but we never tester earlier timepoints) and I wonder if the authors tried to perform a kinetic analysis in particular of earlier timepoints, to determine the time it would take, e.g. after adoptive transfer, before the NK cells retune? If an NK cell that had been retuned (e.g. RAE-ε KO NK cell injected into wt mice) were taken out from the new environment again and cultured in vitro, how long time would it take before NKG2D levels would again rise?

On a more philosophical note, while it is very nice to see evidence for a tuning interaction between an activating receptor and its ligand in vitro, it is not obvious why this interaction should be restricted to HEV in lymph nodes. The argument that NK cells pass thorough lymph nodes is fine, but they also pass through the spleen (probably to a greater extent) and yet spleen endothelial cells do not seem to tune NK cells in the same way. What would be the reason for this?

The data also opens up these and many additional new avenues for further research and I wish to congratulate the authors on a very nice discovery.

---

## [Author Response]

Essential revisions:The reviewers and Reviewing Editor have viewed this manuscript favorably but there are some issues that need attention, specifically:1) Provide in vitro cytotoxicity assays to verify the degranulation assays as recommended by two reviewers.

We thank the reviewers for this helpful suggestion. We performed standard 4- hour ^51^Cr cytotoxicity assays using peritoneal wash cells from WT, RAE-1-KO, and NKG2D-KO mice as effectors against YAC-1 target cells. RAE-1-KO effector cells killed YAC-1 targets significantly more efficiently than WT cells, whereas NKG2D-KO cells killed YAC-1 cells less efficiently than WT cells (Figure 2—figure supplement 1).

We carried out another, related, in vitro experiment in which we directly compared NK responses of WT, RAE-1-KO, and NKG2D-KO cells to stimulation with plate-bound anti-NKG2D antibodies (Figure 3—figure supplement 1): as in the cytotoxicity experiment, there was an elevated response of NK cells from RAE-1-KO mice but not of NK cells from NKG2D KO mice (as they lack the receptor for stimulation).

These data collectively are related to reviewer 2’s comment #4. The key issue in interpreting the tumor killing/rejection experiments is the extent to which the tumor killing is (1) promoted due to direct targeting of tumor cells by NKG2D vs. (2) diminished by desensitization induced by host NKG2D-RAE-1 interactions. In published studies from our group and others, the killing of tumor cells expressing NKG2D ligands is decreased with NKG2D-KO NK cells as compared to WT NK cells, and we describe the same here for B16-MULT1 and YAC-1 cells (Figure 6 and Figure 2—figure supplement 1). Therefore, the powerful pro-killing effect of direct targeting of tumor cells by NKG2D outweighs the desensitization resulting from NKG2D-host RAE-1 interactions in these contexts. Thus NKG2D-mediated killing can mediate immunosurveillance. In contrast, for tumors that are killed by NK cells but not directly targeted by NKG2D, the desensitizing effect of NKG2D- host RAE-1 interactions causes impairment of tumor killing. Thus, deletion of host RAE-1 reverses NK desensitization while preserving NKG2D targeting of tumor cells, resulting in substantially better killing/rejection of NKG2D-ligand expressing tumor cells by NK cells in RAE-1 KO mice.

These issues are now discussed in much greater detail in the Discussion as requested by the reviewer.

2) Expression of NKG2D on NK cells from the BM and liver should be presented.

We have performed these experiments and included them in Figure 1—figure supplement 2. We were surprised that resident liver NK cells are also engaged by host RAE-1 molecules, given that we did not detect RAE-1ε expression in the liver. We first considered the hypothesis that RAE-1ε was present in the bloodstream, e.g., in exosome form, but serum from WT mice failed to downregulate NKG2D on NK cells from RAE-1-KO cells (Author response image 1). We do not know the source of RAE-1ε responsible for engaging resident liver NK cells; one hypothesis is that resident liver cells traffic to local but anatomically distinct structures containing RAE-1ε -expressing cells, perhaps liver-specific lymphoid tissue. We have added these considerations to the Discussion section.

**Author response image 1. respfig1:** NK cells from WT or RAE-1-KO mice were incubated with undiluted serum from WT or RAE-1- KO mice at 37°C for one hour, and NKG2D levels were analyzed.

3) Address reproducibility of BM chimera data.

We have added this data, showing replication in multiple animals, to Figure 4—figure supplement 2.